# Correlation Between the Clinical and Histopathological Results in Experimental Sciatic Nerve Defect Surgery

**DOI:** 10.3390/medicina61020317

**Published:** 2025-02-11

**Authors:** Andrei Marin, Vlad Herlea, Alice Bancu, Carmen Giuglea, Dana Antonia Țăpoi, Ana Maria Ciongariu, Georgiana Gabriela Marin, Silviu Adrian Marinescu, Nicoleta Amalia Dobrete, Adrian Vasile Dumitru, Cristian Trambitaș, Dragoș Șerban, Maria Sajin

**Affiliations:** 1Plastic Surgery Department, St. John’s Hospital, Carol Davila University, 042122 Bucharest, Romania; andrei.marin@umfcd.ro (A.M.); carmen.giuglea@umfcd.ro (C.G.); 2Pathology Department, “Fundeni” Hospital, Carol Davila University, Fundeni Street, 258, 022328 Bucharest, Romania; 3Pathology Department, Sante Clinic, 060754 Bucharest, Romania; alice.bancu@yahoo.com; 4Pathology Department, University Emergency Hospital, Carol Davila University, 050474 Bucharest, Romania; dana-antonia.tapoi@drd.umfcd.ro (D.A.Ț.); ana-maria.ciongariu@drd.umfcd.ro (A.M.C.); vasile.dumitru@umfcd.ro (A.V.D.); maria_sajin@yahoo.com (M.S.); 5Cardiology Department, Oncology Institute, 022328 Bucharest, Romania; gabrielageorgiana88@yahoo.com; 6Plastic Surgery Department, “Bagdasar Arseni” Hospital, 041915 Bucharest, Romania; silviu.marinescu@umfcd.ro; 7Hematology Department, County Emergency Hospital Ploiești, 100137 Ploiești, Romania; amalianicoleta87@yahoo.com; 8Plastic Surgery Department, G. E. Palade University of Medicine, Pharmacy, Science and Technology from Târgu Mureș, 540142 Târgu Mureș, Romania; drtrambitas@gmail.com; 9Surgery Department, University Emergency Hospital, “Carol Davila” University of Medicine and Pharmacy, 020021 Bucharest, Romania; dragos.serban@umfcd.ro

**Keywords:** nerve regeneration, PRP, stem cells, histopathology, experimental model

## Abstract

*Background and Objectives*: Peripheral nerve defect regeneration is subject to ongoing research regarding the use of conduits associated with various cells or molecules. This article aims to correlate histopathological and clinical outcomes at the end of a 12-week experiment performed on a rat sciatic nerve model and show which repair method has the best results. *Materials and Methods*: Forty male Wistar rats were divided into four groups to compare the results of four different methods of reconstruction for sciatic nerve defect: (1) nerve graft–control group, (2) empty aortic conduit, (3) aortic conduit filled with platelet-rich plasma (PRP) and (4) aortic conduit filled with mesenchymal stem cells. There were three clinical examinations: a sensitivity test, a mobility test and a footprint test. After 12 weeks, the nerves were excised and assessed microscopically using conventional Hematoxylin and Eosin staining (HE), special stains and immunohistochemistry (IHC). *Results*: Nerve regeneration was observed in all batches, both from the clinical and histopathological assessment; the two types of examinations correlated for each batch. Immunohistochemistry and special staining offered a more complete image of the nerve regeneration results. *Conclusions*: Superior nerve regeneration was achieved using an aortic conduit in combination with either PRP or stem cells, while the empty aortic conduit recorded lesser results.

## 1. Introduction

Injured nerves require a longer time to regenerate, due to Wallerian degeneration of the axons [1,2,3]. After an initial lesion, the nerves begin to grow at a specific speed of 1 mm/day [4,5]. In order to aid this process, manufactured tubes can be used for nerve gaps up to 70 mm [6]. Furthermore, there has been significant technological development in this field—biocompatible 3D-printed devices or textile conduits can now be used to bridge nerve gaps [7,8]. Generally, nerve conduits offer a good alternative to grafts, alone or in combination with different tissues or substances (stem cells and PRP) [9,10,11].

Nerve regeneration can be properly assessed by clinical evaluation, imagistic investigations [12] and histopathological examination. While clinical evaluation may be difficult to perform, the pathological examination (gross, as well as microscopic) can become a useful tool for evaluating nerve regeneration.

There are three types of nerve injuries in the Seddon classification (neurapraxia, axonotmesis, neurotmesis), while Sunderland uses a 5-degree injury classification [13,14].

The regeneration process after Wallerian degeneration requires a favorable environment as well as a “clear way” along which the axonal sprouts can grow. The Schwann cells, differentiated into Büngner bands, are responsible for directing the axonal growth [15].

There are several new molecules or drugs that aid nerve regeneration: growth factors, PRP and stem cells [16,17,18,19], as well as novel synthetic conduits capable of directing the axonal growth of the nerve. These modern techniques have the potential to replace nerve grafts [20,21,22].

Depending on the level of injury, the axonal growth requires different periods of time for regeneration [23,24,25]. The more proximal the lesion, the longer it will take for the nerve to grow—this nerve growth rate is estimated to be 1 mm/day in humans and 1.5–2 mm/day in rats.

There are several methods of analyzing nerve regeneration in the setting of a previous injury. Clinical evaluation involves assessing both motor and sensitive functions. The motor function can be evaluated using a footprint test and calculating a sciatic functional index (SFI).

The quality of the nerve growth can also be evaluated by directly assessing the nerve at the level of injury. Apart from the conventional histopathology analysis, there are several other valuable immunohistochemistry (IHC) markers that can be used in both qualitative and quantitative studies.

Reconstruction after nerve defects does not have an ideal solution and therefore multiple options should be considered and tested in order to discover which one offers the best results.

## 2. Materials and Methods

### 2.1. Ethical Acknowledgement and Animal Lot Description

Forty male Wistar rats weighing 250–300 g and aged between 55–60 days were divided into 4 groups to assess 4 different techniques used for the repair of a 0.5 cm iatrogenic nerve defect on the right lower limb: (1) nerve graft–control group, (2) empty aortic conduit, (3) aortic conduit filled with platelet-rich plasma (PRP) and (4) aortic conduit filled with mesenchymal stem cells (MSC cell line). Another 2 rats with the same characteristics were sacrificed to obtain the aortic conduits and the PRP for the operations.

The minimal number of animals was used and postoperative analgesia was given to reduce animal suffering. The international protocols for euthanasia were also respected.

Animal care and use were performed according to the laws regarding the well-being of the animal and the project was approved by the Local Ethics Committee (approval no. 1738/17.04.2019). The experiment was conducted according to international guidelines and regulations.

The animals were provided proper lodging in standard cages and fed with normative murine food and water ad libitum. The acclimatization conditions involved 3 spacious rooms dedicated to this experiment, sheltered from outside noise. All rats benefitted from the same lodging conditions, same food and same climate.

### 2.2. Operation and Postoperative Protocol

The rats were anesthetized using a mixture of ketamine and xylazine (75 mg/kg and 10 mg/kg) injected intraperitoneally. All rats were operated on the right lower limb under the same circumstances (microscopic magnification, sterile conditions) and a number of 4–5 stitches were performed for each anastomosis.

All rats received the same anesthesia from a veterinarian and were operated on by the same plastic surgeon, assisted by a colleague to ensure sterile conditions.

The postoperative care implied enroxil 0.003 mg/kg s.c. and meloxicam 1 mg/kg s.c. for 3 days. The operated rats were placed in separate cages for a few days to avoid cannibalism. Body temperature as well as clinical recovery was monitored after each operation for 3 h. The wounds were treated locally in case of dehiscence with Bacitracin/Neomycin ointment. All rats were assessed for 12 weeks postoperative to observe the nerve regeneration.

### 2.3. Follow-Up Protocol

The team that monitored the rats after the operation involved 2 veterinarians, the surgeon and the assistant who operated the rats and several students who provided logistic support. The clinical evaluations were performed by the assistant, while the nerve dissection and nerve excision were performed at the end of 12 weeks after rat euthanasia by the surgeon. The footprint test and the SFI were calculated by a medical engineer. The histopathology and IHC were performed by a team of pathologists. All data were recorded and interpreted by a statistician.

Two rats in the nerve graft group (1) were excluded, one due to postoperative death (the dead rat was then used for aorta harvesting for the second batch) and the other one due to nerve rupture.

#### 2.3.1. Clinical Evaluation

The remaining 38 rats were assessed clinically every other week in terms of sensitivity (pinch test), mobility and footprint test.

The sensitivity test was graded on a scale from 0–3 as follows:No reaction at any level—0 points;Pinch at calf level determines retraction of the limb—1 point;Pinch at ankle level determines retraction of the limb—2 points;Pinch at metatarsal level determines retraction of the limb—3 points.

The mobility test had a scale of 0–3 points:
No movement—0 points;Minimal movement/flexion—1 point;Abduction of the fingers—2 points;Abduction and extension of the fingers–3 points.

The footprint test was performed after the rat passed onto a sponge filled with ink, leaving behind on a graded piece of paper a line of footprints.

The SFI after the Bain and MacKinnon formula was then calculated for each trial.

This index takes into consideration 3 parameters: plantar length (PL) (between the calcaneus and the longest finger), intermediary interdigital length (IT) (between digits 2 and 5) and maximal interdigital length (TS) (between the 1st and 5th digit) [26]. The calculus of this index takes into account the operated as well as the non-operated limb, using the formula (the operated are with E, non-operated with N in front of the above-mentioned parameters):SFI = −38.3 × EPL-NPL/NPL + 109.5 × ETS-NTS/NTS + 13.3 × EIT-NIT/NIT − 8.8

This formula attributes a null value for the uninjured limb and −100 for total loss of function.

#### 2.3.2. Paraclinic Evaluation

At 12 weeks, after the final clinical evaluation, the euthanasia of the rats was performed using a mixture of 225 mg/kg ketamine and 30 mg/kg xylazine administered intraperitoneally followed by an intracardiac injection of 80 mg/kg of KCl. The consequent hyperkaliemia-induced arrhythmia caused cardiac failure and exitus [27].

After the euthanasia, the right sciatic nerve was dissected and examined.

The macroscopic examination at the anastomosis site was based on 2 characteristics:Diameter at the anastomotic site: normal (1 point), narrow/reduced diameter (0 points);Aspect of the nerve: opaque (1 point), translucent (0 points).

After gross examination, the nerves were excised and assessed microscopically using conventional Hematoxylin and Eosin staining (HE), special stains and immunohistochemistry (IHC).

The nerve regeneration was graded based on HE on a scale from 0–3 points accordingly:No regeneration (ruptured nerve)—0 points;Less than 50% normal nerve cells/many visible gaps between the nerve fibers—1 point;50–75% normal nerve cells and few visible gaps between the nerve fibers—2 points;>75% of the normal and small gaps between the nerve fibers—3 points.

The data were analyzed using IBM SPSS Statistics 26. To establish a correlation between the histopathology and clinical findings (taking into consideration the type of variables-nominal and continuous), Cramer’s V. correlation coefficient and Pearson’s correlation coefficient were used.

The nerves were also evaluated using immunohistochemistry and special stains. The following antibodies were used for immunolabeling: S100 (4C4.9 clone, Cell Marque, Rocklin, CA, USA), PGP9.5 (RP clone, Cell Marque), CD34 (QBEnd/10 clone, Cell Marque) and Ku80 (EPR3468 clone, Abcam, Cambridge, UK). The special stains included Masson’s trichrome stain (Diapath, Martinengo, Italy) and Gömori silver stain (Bio-Optica, Milan, Italy).

## 3. Results

Variable nerve regeneration was observed in all batches, both from the clinical and histopathological perspectives. There have been positive statistical correlations, showing robust nerve healing in batches (3) and (4), good healing in batch (1) and lesser results in batch (2).

### 3.1. Macroscopic Examination

The pictures depict the gross examination of the sciatic nerves 3 months after the operation (photos were taken in situ). Some nerves presented a robust, opaque texture, while others had visible narrowing or translucent appearance (Figure 1).

### 3.2. Microscopic Examination

Microscopically, the nerve regeneration was assessed using conventional HE staining, special stains and IHC, shown in the following images (Figure 2, Figure 3, Figure 4, Figure 5, Figure 6 and Figure 7).

#### 3.2.1. HE Stain

On HE, all batches showed variable nerve regeneration. Groups 2–4 showed some periarterial expansion. Batch 2 displayed moderate nervous growth, but the intraluminal fibers exhibited frequent vacuolizations and areas of atrophy.

#### 3.2.2. IHC

The cytoplasmic expression of PGP 9.5 was seen in all 4 groups, with a higher intensity and a broader distribution in batches 3 and 4. Moreover, in batch 2, the staining showed moderate intensity and displayed a heterogeneous pattern. S100 expression paralleled that of PGP 9.5.

In batch 3, CD34 highlighted numerous newly formed capillaries, proving that there was a rich blood supply. In batch 4, it was focally positive in the stellate mesenchymal cells, highly suggestive of their stem phenotype. In batches 2–4, the aortic conduit showed no staining due to endothelial denudation.

Batch 4 showed Ku80 loss.

#### 3.2.3. Special Stains

Masson’s trichrome stain underlined prominent perineural fibrosis in batch 2 while the other groups showed variable collagen deposition around the suture site. In terms of myelin sheaths, batches 1 and 2 displayed faint staining, whereas the staining in batches 3 and 4 was more intense, although discontinuous.

Gömori silver stain showed distorted reticulin meshwork in batch 2, due to prominent intercellular edema. In batch 3, numerous reticulin fibers were seen around the newly formed capillaries. Batch 4 showed delicate reticulin fibers extending into the loose mesenchymal tissue as scaffolds for the sprouting nerve fibers and capillaries.

### 3.3. Statistical Analysis

From a clinical point of view, all 38 rats had a full recovery in terms of sensitivity at 12 weeks (noted with 3 points). Therefore, the correlation between the histopathology and the sensitivity test was achieved using the sensitivity test performed at 10 weeks. For the other two clinical indicators (SFI and motor test), the values obtained in the last test (at 12 weeks) were used for the correlation with the histopathology tests.

The statistical correlations between the clinical and pathological parameters are as follows:

There was no correlation between the macroscopic examination and the sensitivity test at 10 weeks (*p* = 0.12).

There was a positive correlation between the macroscopic examination and the motor test at 12 weeks (*p* = 0.038).

There was a positive correlation between the macroscopic examination and the SFI at 12 weeks (*p* = 0.001).

There was a positive correlation between the microscopic examination and the sensitivity test at 10 weeks (*p* = 0.031).

There was a positive correlation between the microscopic examination and the motor test at 12 weeks (*p* = 0.002).

There was a positive correlation between the microscopic examination and the SFI at 12 weeks (*p* < 0.001).

There was a positive correlation between the micro- and macroscopic examinations (*p* < 0.001).

The histology image of the nerve with nerve fibers occupying >75% of the sectioned nerve with small gaps between the nerve fibers (a microscopic indicator of a robust nerve regeneration) correlated with good results in all three of the tests (sensitivity test at 12 weeks, motor test at 12 weeks and SFI at 12 weeks). In contrast, fibers occupying <25% of the nerve circumference with large gaps in between (an indicator of poor nerve regeneration) correlated with overall poor test results regarding sensitivity, motor function and SFI.

## 4. Discussion

The positive correlations (macroscopic examination and the sensitivity test; macroscopic examination and the SFI; microscopic examination and the sensitivity test; microscopic examination and the motor test; microscopic examination and the SFI) indicate that histopathology findings (both microscopic and macroscopic) are tightly connected with the results of the clinical evaluation at the end of the experiment. Histopathology and immunohistochemistry are valuable investigations that can be used for ex vivo evaluations of nerve regeneration.

Microsurgery requires skills and training in order to perform correct nerve anastomosis. Proper preoperative preparation and a good surgical technique are important to prevent complications. Chlorhexidine is a potent disinfectant, which can be used at the surgical site with maximal efficiency to prevent infections, while wound dehiscence could be treated in specific cases conservatory [28,29,30,31].

Postoperative care has also great significance for the outcome. Meloxicam prevents inflammation, reducing pain in rats [32]. Blythe et al. have used the rat grimace scale to evaluate the effects of Meloxicam on neuropathic cervical pain caused by cervical root compression and demonstrated that NSAID reduces this pain [33].

In terms of clinical examination, the SFI has been a standard measurement for evaluating nerve regeneration after different types of sciatic nerve injury. However, due to its possible subjectivity, new methods such as video gait analysis and quantitative measurement of isometric tetanic muscle force have been suggested [34].

Aside from the pinch test, Wang et al. used novel assessment techniques: laser Doppler perfusion imaging, walking track analysis and transmission electron microscopy (the latter being able to show unmyelinated axons in the injured sciatic nerve) [35].

Extensive work on peripheral nerve defect repair in rats was performed by Siemionow et al. In her studies, she evaluated the rats using functional (pinprick, toe-spread), neurosensory (somatosensory-evoked potentials) and histomorphometric evaluations. Siemionow evaluated the results at 3, 6 and 12 weeks [36]. In the same study, the authors showed that the epineural sheath graft technique (a flat rectangular epineural sheath created after removing the fascicles from the excised nerve) was a good method for nerve regeneration [37]. Another study conducted by Siemionow et al. showed that epineural tubes containing isogenic bone marrow stromal cells (BMSC) could also be a good alternative to autograft repair [38].

One more study conducted by Nijhuis et al. used the toe spread and pinprick test to evaluate motor and sensory function for a 15 mm sciatic nerve defect repaired in three different ways: nerve autograft (group 1), a vein filled with muscle and bone marrow stromal cells BMSC (group 2) and a vein filled just with muscle (group 3). In this study, group 2 presented more Schwann cells compared to group 3, proving that BMSC have a beneficial effect when used in nerve conduits [39]. Another study conducted by Nijhuis showed that a vein graft supported with isogenic bone marrow stromal cells provides better nerve regeneration for 20 mm nerve gaps in comparison to an empty venous graft or a vein graft filled with saline [40].

Nerve allografts from cadavers have also been considered for nerve defects; however, these are rapidly rejected if proper immunosuppression is not achieved [41].

Aside from clinical tests, histopathological examination employing both classical HE and special stains, as well as novel immunohistochemistry markers, can tremendously aid in regeneration quantification.

Morphologic features that correlated with better nerve regeneration on HE staining were intact myelin sheaths, thick continuous fibers, little to no nerve vacuolizations and intercellular edema, newly formed capillaries, minimal perineural fibrosis, an optimal amount of reticulin and collagen meshwork.

The correlations between clinical and histological outcomes showed that the opaque robust nerve texture (a macroscopic indicator of optimal nerve regeneration) correlated with good results of the motor test and of the SFI at 12 weeks, but not with the optimal results of the sensitivity 10-week test. In opposition, the translucent aspect of a narrow nerve (a macroscopic indicator of poor nerve regeneration) correlated with poor motor test and SFI results but did not correlate with the results of the sensitivity test.

Similar to our research, another study conducted by Pavic et al. used immunohistochemical methods in a sciatic nerve crush injury model. They showed changes in sensory and motor axons in the spinal cord segment L3–L6 on the injured side, corresponding to the plantar test results, by using antibodies for Myelin-associated glycoprotein (MAG) and gangliosides GD1a and GT1b on the aforesaid part of the spinal cord [42].

Immunohistochemistry was used in our study to increase the accuracy of the evaluation.

PGP 9.5 is a general neural marker. A positive reaction with anti-PGP9.5 antibodies is seen in neurons, nerve fibers and neuroendocrine cells [43,44,45]. In our study, the intensity and cytoplasmic distribution of PGP 9.5 were proportional to nerve regeneration. On one hand, the heterogeneous pattern of PGP 9.5 in batches 1 and 2 was consistent with focal nerve damage. On the other hand, the higher intensity and the broader distribution of PGP 9.5 expression in batches 3 and 4 indicated better nerve regeneration in these two groups.

S100 represents a calcium-binding cytosolic protein family. Neurons, Schwann cells and dendritic cells exhibit S100 positivity. In damaged nerve fibers, the staining might be decreased or even absent [46,47]. In our study, its expression paralleled that of PGP9.5, showing better regeneration in batches 3 and 4 and patchy positivity, with varying intensity, in batches 1 and 2.

CD34 is a transmembrane glycoprotein expressed in a variety of cells. A positive reaction with anti-CD34 antibody is seen in stem cells and endothelial cells in a membranous pattern [48]. The positivity of CD34 in batch 3 indicated that the PRP used was effective in generating new blood vessels, therefore improving overall nerve regeneration. Although it was expected for CD34 to be positive in the batches in which an aortic conduit was used, the negative staining in these cases could be attributed to the denuded endothelium and its degenerative changes.

Ku80 is a nuclear protein that promotes the repair of double-strand DNA breaks. Its absence is correlated with tumorigenesis and premature degenerative processes [49,50,51]. Ku80 was used in batch 4 to assess the quality of nerve regeneration. Although the nerve fibers were morphologically intact in batch 4, the loss of Ku80 staining suggests that DNA repair might be impaired in the long run.

Special stains were used to better assess the architecture and associated collagen and reticulin meshworks.

Gömori silver stain highlights the reticulin fibers, which are argyrophilic and therefore appear black [52]. In our study, the reticulin meshwork was prominent in batches 3 and 4, directly proportional to the nerve regeneration. By contrast, batches 1 and 2 displayed distorted reticulin fibers due to poorer nerve regeneration.

Masson’s trichrome stain is a special stain that highlights connective tissue fibers. In normal peripheral nerves, the associated collagen fibers of the epi-, peri- and endoneurium appear blue, whereas the myelin sheaths stain purple–red. In the case of myelin sheath damage, the stain becomes faint or is absent [53,54,55]. In our study, there was considerable perineural fibrosis in batches 1 and 2. This finding was associated with inferior nerve regeneration. Myelin sheath staining was more intense in batches 3 and 4, a feature that correlated with optimal nervous growth.

In another study with transected rat sciatic nerves, some repaired with tubular collagen nerve sleeves, others without, the histologic examination of nerve regeneration observed three aspects: neuroma formation, connective tissue proliferation and axonal regrowth. When comparing the wrapped to non-wrapped repaired nerves, Kim et al. showed that significantly more connective tissue formation was present in non-wrapped nerves, but there was no neuroma formation and no significant difference in axonal growth [56].

Evaluation of nerve regeneration can be performed using different techniques: in situ high-resolution ultrasound (HRUS) magnetic resonance microscopy (MRM), histological cross-sections (HCS) and optical projection tomography (OPT) [57].

Furthermore, stem cells constitute an innovative solution not only for peripheral nerve regeneration but also for obtaining functional neurons in vitro. Soni A. et al. have developed an efficient method for converting neural progenitor cells into functional neurons, which are then available without further animal sacrifice [58].

Regarding the loss of Ku80 protein expression by immunohistochemistry in batch 4, it would be interesting to explore this phenomenon at the molecular level—some studies indicate that Wnt and COX2 signaling pathways are involved in its up- and downregulation in nerve defects inflicted in mice. As electron microscopy and immunofluorescence studies are required to further investigate this hypothesis, we hope to look into it in a future experimental setting [59,60,61].

A limitation of our study is that the results after PRP and stem cell treatment could have been better in clinical evaluation compared to batch 2, but more rats should have been included in the study for a significant statistical result.

## 5. Conclusions

Nerve regeneration was observed in all batches, both from the clinical and histopathological perspectives. Positive correlations between the clinical and the histopathological evaluations were observed in the study. Promising results were obtained in batches where PRP and stem cells were used, followed by nerve autograft.

Superior nerve growth correlated with diffuse and intense expression of the neural markers PGP9.5 and S100. The vascular marker CD34 was expressed in batches 3 and 4, due to rich neovascularization, which was also linked to better nervous regeneration. Denser perineural fibrosis and poorer nerve regeneration were seen in batch 2. The markers used in the last batch showed excellent nerve regeneration.

In conclusion, immunohistochemistry and special staining are useful tools in the assessment of nerve regeneration and could be corroborated with other investigations in experimental projects.

## Figures and Tables

**Figure 1 medicina-61-00317-f001:**
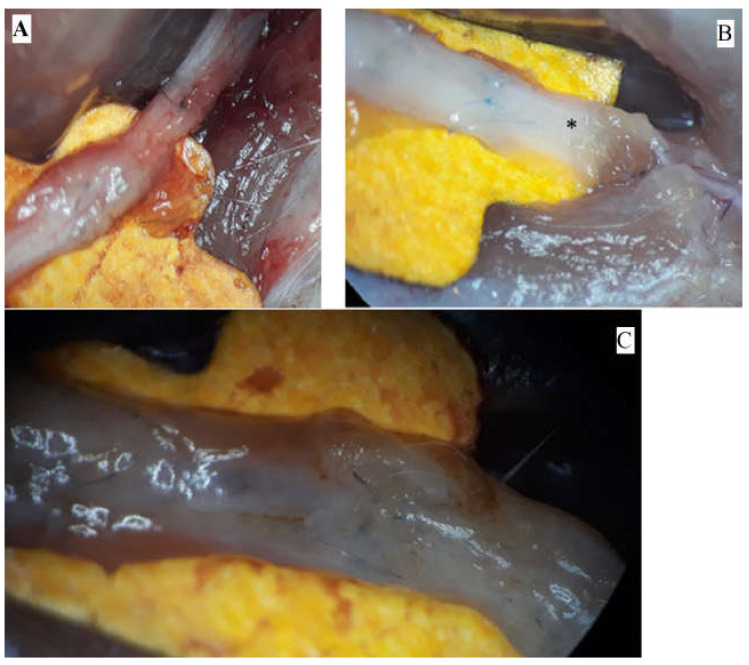
Macroscopic aspects of nerves after regeneration. (**A**) Narrow sciatic nerve-auto-graft site. (**B**) See-through sciatic nerve in the distal end (marked *). (**C**) Robust, opaque sciatic nerve at the site of the nerve repair.

**Figure 2 medicina-61-00317-f002:**
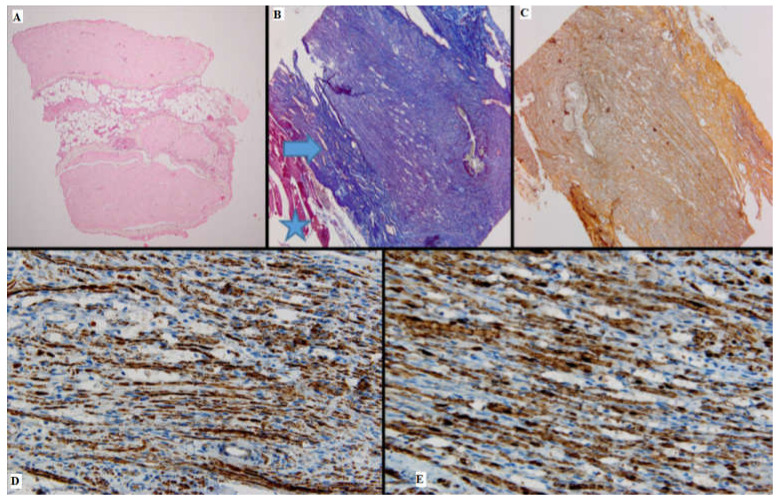
Batch 1, proximal level: (**A**) Normal nerve bundles—longitudinal and transverse sections (HE, 4×); (**B**) Masson’s trichrome stain: There are striated muscle peripherally (star), the connective tissue of the perineurium (arrow) and nerve fibers with attenuated, pale purple–red myelin sheaths. The suture is seen on the right side of the image (10×); (**C**) Gömori stain: fine reticulin fibers in a longitudinal plane of section (10×); (**D**) PGP 9.5 immunostain is strongly positive in a cytoplasmic pattern in most of the nerve fibers (10×); (**E**) S100 stain: strong nuclear and cytoplasmic positivity in the majority of the nerve fibers; moderate intercellular edema (40×).

**Figure 3 medicina-61-00317-f003:**
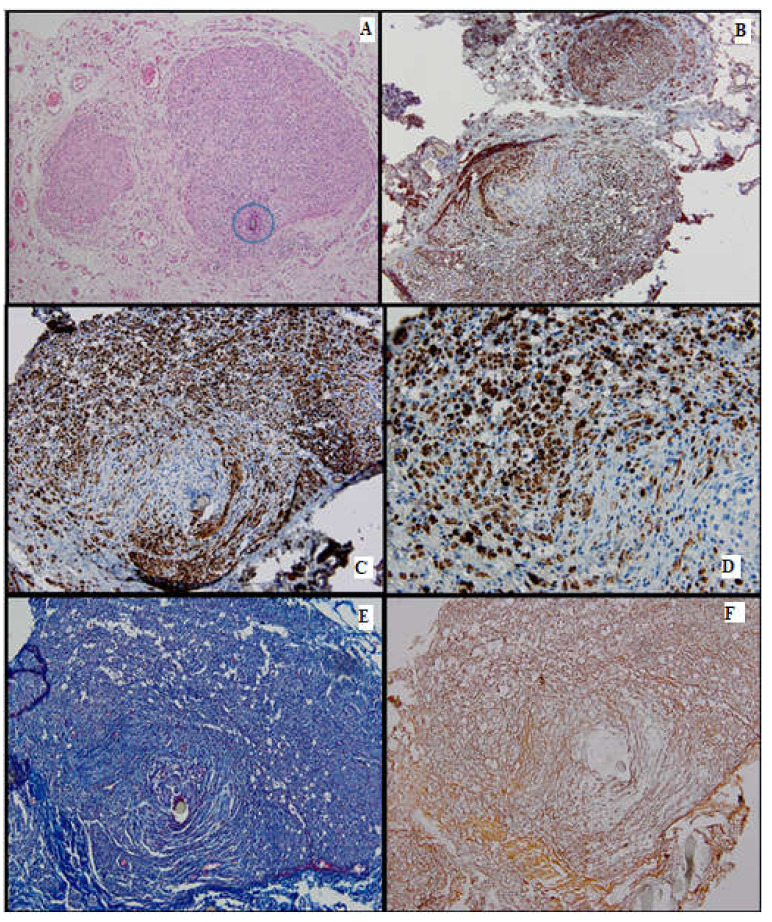
Batch 1, distal level: (**A**) HE-moderate nervous regeneration; intra- and intercellular edema are readily apparent (10×); the circle highlights the suture site (**B**) PGP 9.5-overall cytoplasmic positivity, with varying intensity among the nerve fibers, in a mosaic pattern (10×); (**C**,**D**) S100-moderate nuclear and cytoplasmic positivity, in a heterogeneous pattern; lack of staining in the damaged nerve fibers (20×–40×); (**E**) Masson’s trichrome stain highlighting the connective tissue meshwork. The myelin sheaths display weak discontinuous staining (20×); (**F**) Gömori stain: prominent intercellular edema; nerve fibers show architectural distortion and attenuated reticulin meshwork (20×).

**Figure 4 medicina-61-00317-f004:**
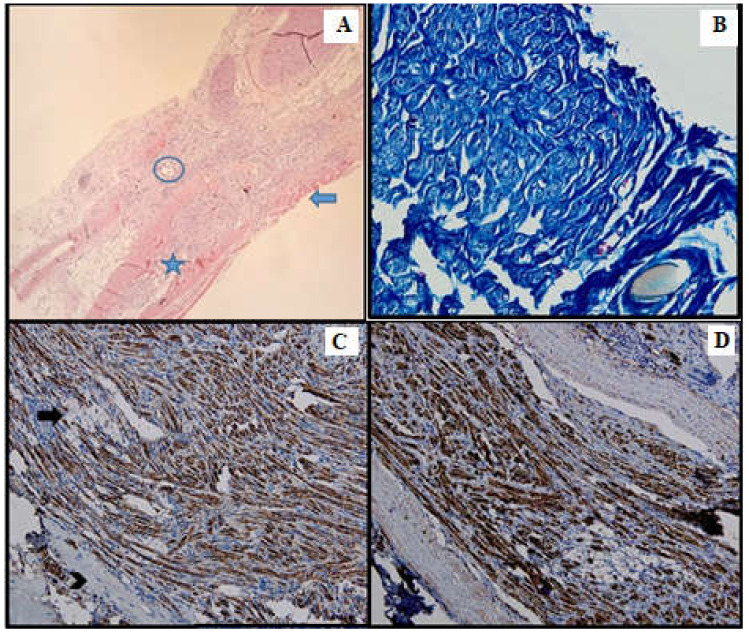
Batch 2, point of junction between proximal and distal levels: (**A**) Atrophic striated muscle (arrow), fibrotic aortic conduit and thin nerve fibers showing weak-to-moderate regeneration (star). The circle highlights the suture (HE, 10×); (**B**) Masson’s trichrome stain: The nerve fibers are disorganized and interrupted, displaying faint myelin sheath staining (40×). (**C**) PGP 9.5 shows moderate cytoplasmic staining. Many fibers are immunonegative and display vacuolizations (arrow). Viable striated muscle is focally visible in the periphery (arrowhead). The suture can be seen in the upper right corner (20×); (**D**) S100 shows moderate nuclear and cytoplasmic positivity (20×).

**Figure 5 medicina-61-00317-f005:**
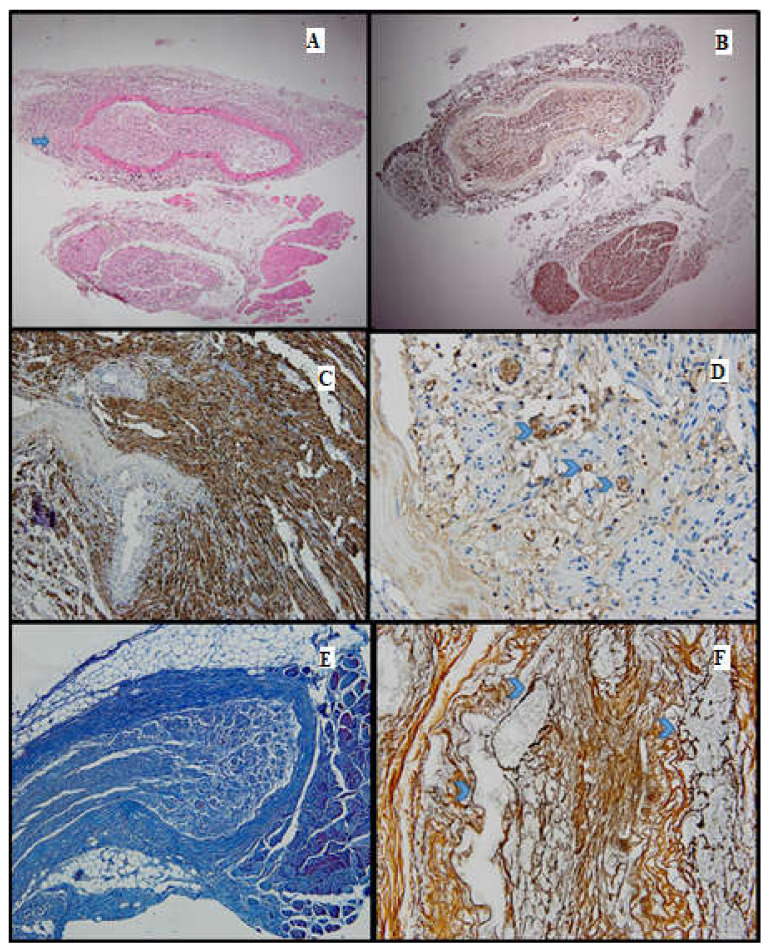
Batch 3, proximal and distal levels: (**A**) HE-proximal level (lower section) and distal level (upper section): Similar nerve calibers and optimal nerve regeneration. Distally there are gaps through which nerve bundles penetrate the surrounding tissue (arrow, 4×); (**B**) proximal and distal levels showing strong PGP9.5 staining and similar caliber (4×); (**C**) S100-intense staining in the majority of the distal nerve fibers and their growth into adjacent tissue (10×); (**D**) CD34-numerous capillaries among the distal nerve fibers (arrowheads, 40×); (**E**) Masson’s trichrome stain highlighting the connective tissue meshwork of the distal region (10×); (**F**) Gömori silver stain highlighting in black the fine reticulin fibers surrounding the capillaries in the distal portion of the nerve (arrowheads, 40×).

**Figure 6 medicina-61-00317-f006:**
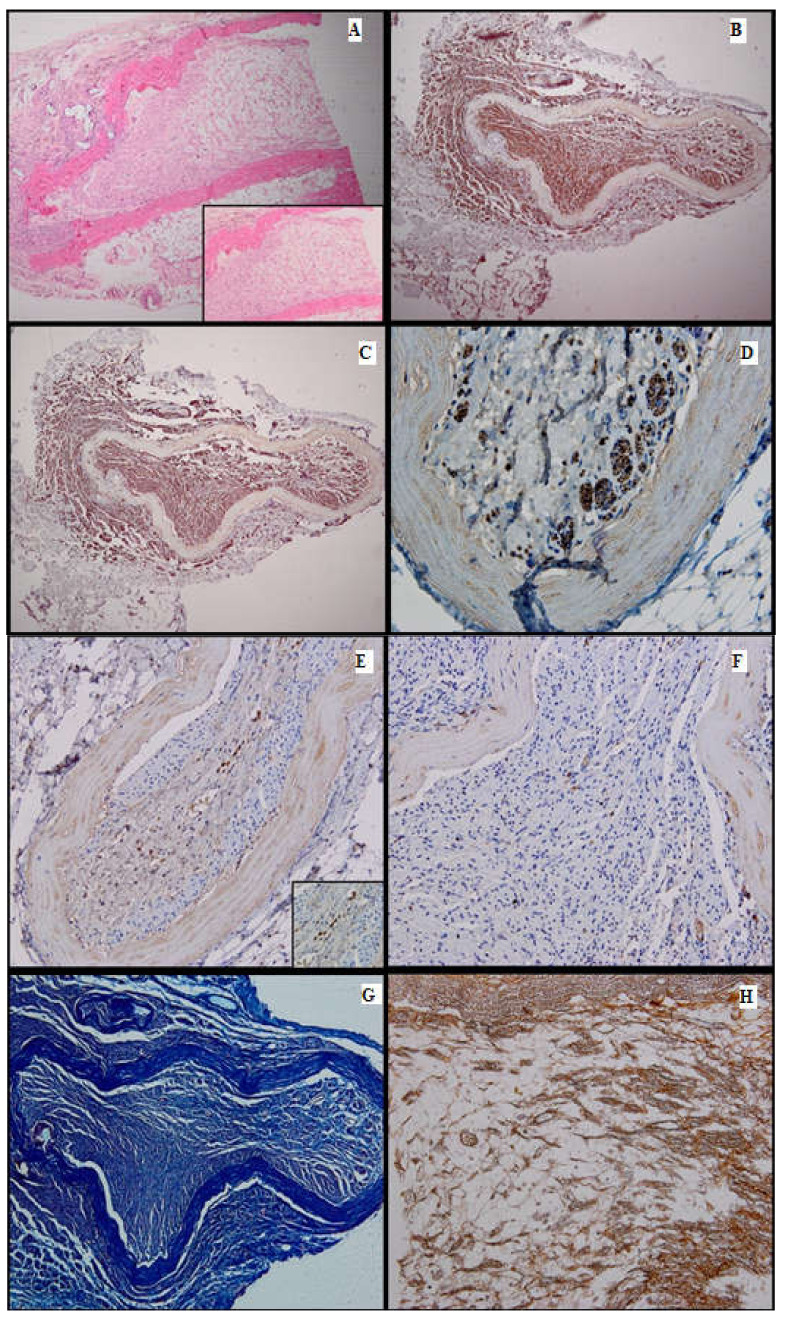
Batch 4, distal level: (**A**) Delicate mesenchymal tissue with regenerating nerve fibers (HE, 4×). Inset shows loose mesenchymal tissue with numerous capillaries and scattered stellate cells displaying a stem phenotype (HE, 10×); (**B**) S100-prominent nerve fiber regeneration; some expansion throughout the aortic conduit and extension into adjacent tissue. There is positive internal control in the peripheral adipose tissue (4×); (**C**) PGP9.5-same growth pattern as seen with S100 immunostain (4×); (**D**) PGP9.5-strong cytoplasmic positivity in the nerve fibers extending into the endoluminal mesenchymal tissue (40×); (**E**) CD34-focal, moderate-to-intense membranous positivity in the stellate cells, highly suggestive of their stem phenotype (20×). Inset displays positive mesenchymal star-shaped cells (40×); (**F**) Ku80 stain is negative in the regenerating portion of the nerve. This finding suggests that DNA repair might be impaired in the long run (20×); (**G**) Masson’s trichrome stain highlights the connective tissue; faint myelin sheath staining in most of the nerve fibers (10×); (**H**) Gömori stain underlines the fine reticulin meshwork extending into the mesenchymal tissue, paralleling the growth of the nerve fibers and capillaries (20×).

**Figure 7 medicina-61-00317-f007:**
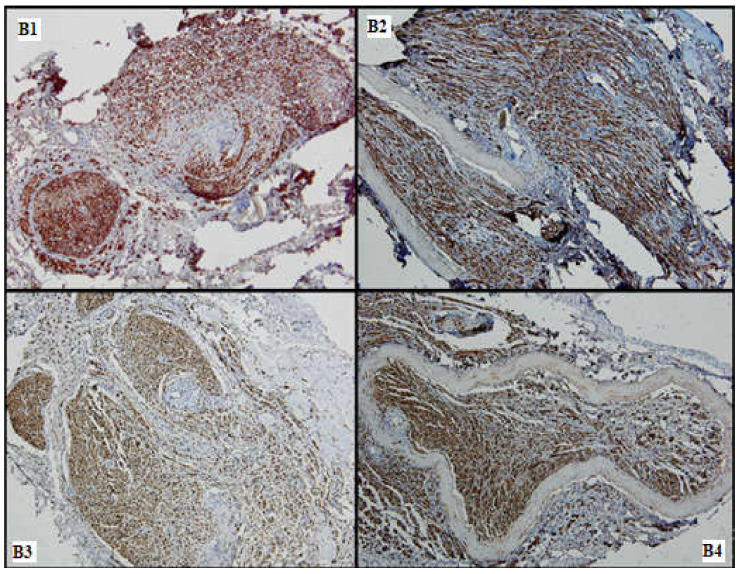
Nervous regeneration in the distal portions of the 4 batches (**B1**–**B4**), comparative aspects: Batches 1, 3 and 4 show superior nervous growth as compared to batch 2; periarterial expansion is a common feature seen in (**B2**–**B4**) (S100, 10×).

## Data Availability

The data is unavailable due to privacy or ethical restrictions.

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
