# Peer review of "Correlation Between the Clinical and Histopathological Results in Experimental Sciatic Nerve Defect Surgery"

_medicina, 2025, doi:10.3390/medicina61020317_

Round 1
Reviewer 1 Report
Comments and Suggestions for Authors
Abstract: The aim is poorly written and is not adequately addressed by the methods. The authors need to rephrase the aim to clearly state the objective of the study. They should articulate what this study seeks to demonstrate. Nerve regeneration, both at the histological and molecular levels, has been extensively studied and does not represent a novelty. While the authors compared different reconstruction techniques, there is no control group included in the study.
Overall, the text is filled with grammatical errors and some scientific inaccuracies. Each sentence within the introduction is supported by up to six references, even for simple statements containing general information—this results in redundant or empty citations. The introduction has significant room for improvement. For example, the inclusion of the SFI formula in the introduction is unnecessary and may confuse the reader. This information is better suited to the methods section if even needed.
Furthermore, the introduction lacks a funnel-shaped structure and instead presents random data, loosely connected to nerve regeneration. It should clearly emphasize the main problem, with a logical progression leading to the aim, which must be explicitly stated in the final paragraph. The aim should align with the authors' protocol and be designed to evaluate a specific hypothesis.
Methods: The methods section should begin with an ethical statement, followed by a concise description of the rodent population and their housing environment, which is currently missing. Key details such as the age, weight, and other basic characteristics of the rats must be included. Figure with protocol must be added (ie, flowchart) to help the reader that glances through.
The methods also lack proper organization. Subsections would make this section more reader-friendly, and tables summarizing the assessments would provide clarity. Additional information is needed about the assessments—who performed them, and what literature supports the evaluation methods?
The figure descriptions are inadequate. For example, Fig. 1 lacks proper textual description, and figure descriptions should not be incorporated into the figure itself. Visible autocorrect underlines can be seen! remove. The labels on figures from Fig. 2 onward are too small to be legible.
The authors should describe in greater detail the correlations presented in section 3.3. I cannot understand what authors did here.
Discussion: The discussion requires significant improvement. Authors should include other articles evaluating nerve integrity with other microscopic techniques. I suggest authors to include findings of the following article and comment how these modalities could aid in peripheral nerve integrity assessment: https://doi.org/10.1038/s41598-024-84396-y Furthermore, there is excessive, irrelevant text, including descriptions of staining techniques and their affinities, which are unrelated to the study's core findings. The discussion should begin by summarizing the main findings, followed by a comparison of these findings with existing literature. It should focus on interpreting the results in the context of the study's aim and hypothesis.
Comments on the Quality of English LanguageEnglish syntax and grammar must be approved.
Author Response
"The aim is poorly written and is not adequately addressed by the methods. The authors need to rephrase the aim to clearly state the objective of the study. They should articulate what this study seeks to demonstrate. Nerve regeneration, both at the histological and molecular levels, has been extensively studied and does not represent a novelty. While the authors compared different reconstruction techniques, there is no control group included in the study."
The abstract has been changed in order to express the aim more clearly. The control group is the first group (the nerve graft group), which is currently the gold standard for nerve reconstruction - I have added this in the text. A control group in which no action would be performed after sciatic nerve injury would be cruelty against animals, as there is no chance of recovery if the nerve is sectioned and has a 0.5cm gap.
"The text is filled with grammatical errors and some scientific inaccuracies."
Most grammar errors have been corrected and the scientific inaccuracies have been analyzed and amended.
"Each sentence within the introduction is supported by up to six references, even for simple statements containing general information—this results in redundant or empty citations."
I have reduced the number of references supporting some of these sentences by detailing each sentence and dividing them.
"inclusion of the SFI formula in the introduction is unnecessary and may confuse the reader. This information is better suited to the methods section if even needed."
I have moved the SFI formula from the introduction and added it in the methods sections. It could be useful for the understanding of the text.
"the introduction lacks a funnel-shaped structure and instead presents random data, loosely connected to nerve regeneration. It should clearly emphasize the main problem, with a logical progression leading to the aim, which must be explicitly stated in the final paragraph. The aim should align with the authors' protocol and be designed to evaluate a specific hypothesis."
I have reshaped the structure of the introduction in order to funnel the ideas to the last paragraph - reconstruction after nerve defects doesn't have a an ideal solution and therefore multiple options should be considered and tested to see which one offers the best results.
"Methods: begin with an ethical statement, followed by a concise description of the rodent population and their housing environment, which is currently missing. Key details such as the age, weight, and other basic characteristics of the rats must be included. Figure with protocol must be added (ie, flowchart) to help the reader that glances through. "
The information about rodent population was added in the text.
"Subsections would make this section more reader-friendly, and tables summarizing the assessments would provide clarity. Additional information is needed about the assessments—who performed them, and what literature supports the evaluation methods?"
Subsections have been added as well as information about the team who worked in this project and literature which supports the evaluation methods. I have also added dr. Siemionow experiments which supports these evaluation methods.
" For example, Fig. 1 lacks proper textual description, and figure descriptions should not be incorporated into the figure itself. Visible autocorrect underlines can be seen! remove. The labels on figures from Fig. 2 onward are too small to be legible."
Fig. 1 was modified. The labels were also changed to be more visible
"The authors should describe in greater detail the correlations presented in section 3.3. I cannot understand what authors did here."
The correlations presented in the results sections are interpreted at the beginning of the discussion section - the main idea is that the histopathologic findings (both microscopic and macroscopic) are connected to the clinical results.
"Authors should include other articles evaluating nerve integrity with other microscopic techniques. I suggest authors to include findings of the following article and comment how these modalities could aid in peripheral nerve integrity assessment: https://doi.org/10.1038/s41598-024-84396-y "
I have added more articles supporting this experiment.
"there is excessive, irrelevant text, including descriptions of staining techniques and their affinities, which are unrelated to the study's core findings. The discussion should begin by summarizing the main findings, followed by a comparison of these findings with existing literature. It should focus on interpreting the results in the context of the study's aim and hypothesis. "
The discussion section has been re-written.
Reviewer 2 Report
Comments and Suggestions for Authors
The article entitled "Correlation between the clinical and histopathologic results in experimental sciatic nerve defect surgery" addresses a significant topic in regenerative medicine, focusing on the repair of peripheral nerve injuries. By examining the efficacy of different repair methods, such as aortic conduits, PRP (platelet-rich plasma), and stem cells, the study provides valuable insights into innovative approaches for nerve regeneration. However, some aspects could be improved to enhance the scientific scope and depth of the study.
Firstly, the study's sample size, involving 40 rats divided into four groups, is relatively small. While it allows for preliminary conclusions, a larger sample size would improve the robustness of statistical analyses and the reliability of the findings. In addition, the rationale for the chosen number of animals and their allocation to groups could have been better detailed.
Secondly, the absence of comprehensive details on randomization and post-operative monitoring represents a methodological limitation. For instance, the study does not clarify whether measures were taken to reduce individual variability, such as controlling for factors like age or weight. Such details are crucial to ensuring the reproducibility of the findings and minimizing bias.
From a clinical perspective, the study establishes an interesting correlation between clinical assessments (e.g., motor function tests) and histopathological observations. However, the functional tests employed, such as the SFI (sciatic function index), could be complemented by more advanced techniques like automated video tracking or electrophysiological assessments to provide a more detailed evaluation of nerve recovery.
Moreover, the study mentions the formation of capillaries and the loss of specific markers such as Ku80 in the stem cell-treated group. While these observations are intriguing, their implications are not fully discussed. A more in-depth analysis of the molecular mechanisms underlying these findings would add significant value. Similarly, the poor outcomes observed in the group treated with the aortic conduit alone (group 2) could benefit from further exploration to identify potential barriers to effective regeneration.
Ethical considerations, while briefly mentioned, could have been elaborated further. For example, a clear explanation of the measures taken to minimize animal suffering and a declaration of adherence to ethical guidelines would enhance the study's credibility.
In terms of generalizability, the exclusive use of a rat model raises questions about the applicability of these findings to humans. While the study makes an important contribution to preclinical research, extending similar experiments to larger animal models, such as primates, would be necessary before translating the results into clinical practice.
Finally, the study's discussion could be enriched by comparing its findings to those of similar research in the field. For example, how do the histopathological and functional outcomes reported here compare to those obtained using alternative conduits or treatment combinations in other studies?
In conclusion, this study makes a valuable contribution to understanding sciatic nerve regeneration and highlights the potential of combining aortic conduits with PRP or stem cells. With methodological refinements, a broader discussion of its limitations, and a focus on long-term outcomes, this research could have an even greater impact in advancing the field of peripheral nerve repair.
Author Response
"Firstly, the study's sample size, involving 40 rats divided into four groups, is relatively small. While it allows for preliminary conclusions, a larger sample size would improve the robustness of statistical analyses and the reliability of the findings. In addition, the rationale for the chosen number of animals and their allocation to groups could have been better detailed."
While I realize that 40 rats in 4 groups is a small number to assess results in terms of statistical results (aspect which is a limitation of this study), the study was designed this way in order to meet the criteria for approval from the local ethical committee, which states that one should use the smallest number of animals for experiments, under the 3R principle- replace (with non-living where possible), reduce (the number of animals), refine (the procedures for minimum suffering for animals).
"Secondly, the absence of comprehensive details on randomization and post-operative monitoring represents a methodological limitation. For instance, the study does not clarify whether measures were taken to reduce individual variability, such as controlling for factors like age or weight. Such details are crucial to ensuring the reproducibility of the findings and minimizing bias."
The details regarding the animals - age, weight, accommodation - have been added in the text.
"However, the functional tests employed, such as the SFI (sciatic function index), could be complemented by more advanced techniques like automated video tracking or electrophysiological assessments to provide a more detailed evaluation of nerve recovery."
Unfortunately the study did not have the logistics for automated video tracking or electrophysiological; however, in the same study, there were other assessments such as MRI for gastrocnemius muscles and gastrocnemius index (after weighing the gastrocnemius muscles) which were presented in another article which was cited -
Marin, A, Savescu, M., Marin, G.G., Dricu, A., Parasca, S., Giuglea, C. Evaluation of muscle atrophy after sciatic nerve defect repair – experimental model. Romanian Journal of Military Medicine. Vol. CXXV • No. 3/2022 doi: 10.55453/rjmm.2022.125.3.10
The present article did not include these assessments, as this would have been too much for one article.
"Moreover, the study mentions the formation of capillaries and the loss of specific markers such as Ku80 in the stem cell-treated group. While these observations are intriguing, their implications are not fully discussed. A more in-depth analysis of the molecular mechanisms underlying these findings would add significant value. "
These paragraphs have been rewritten and mire information about the molecular mechanisms added.
"Ethical considerations, while briefly mentioned, could have been elaborated further. For example, a clear explanation of the measures taken to minimize animal suffering and a declaration of adherence to ethical guidelines would enhance the study's credibility."
The minimal number of animals was used and postoperative analgesia was given to reduce animal suffering. The international protocols for euthanasia were also respected. We have added this in the text.
"In terms of generalizability, the exclusive use of a rat model raises questions about the applicability of these findings to humans. While the study makes an important contribution to preclinical research, extending similar experiments to larger animal models, such as primates, would be necessary before translating the results into clinical practice."
For any study, experiments on larger animals should be performed. This study constitutes a primary experiment to evaluate different nerve gap repairs.
"Finally, the study's discussion could be enriched by comparing its findings to those of similar research in the field. For example, how do the histopathological and functional outcomes reported here compare to those obtained using alternative conduits or treatment combinations in other studies?"
We have added some other similar research and discussed further the utility of these treatments.
Round 2
Reviewer 1 Report
Comments and Suggestions for Authors
The authors addressed some of my previous concerns and slightly improved the manuscript. I encourage authors to proofread the manuscript and again, improve the rest of the proposed parts, particularly introduction is weak part that should be proofread and improved in content as well as in english syntax.
Comments on the Quality of English LanguageShould be improved or at least improved with some free online tool for proofreading.
Author Response
The authors addressed some of my previous concerns and slightly improved the manuscript. I encourage authors to proofread the manuscript and again, improve the rest of the proposed parts, particularly the introduction is the weak part that should be proofread and improved in content as well as in English syntax.
Thank you for your feedback. The article was checked again using Grammarly and proofread by two native English speakers. The introduction was amended as solicited and the grammar errors were corrected.